# SNODE: Spectral Discretization of Neural ODEs for System Identification

**Alessio Quaglino,**[*]   **Marco Gallieri,**   **Jonathan Masci,**   **Jan Koutník**

NNAISENSE, Lugano, Switzerland

`{alessio, marco, jonathan, jan}@nnaisense.com`

## Abstract

This paper proposes the use of spectral element methods (Canuto et al., 1988) for fast and accurate training of Neural Ordinary Differential Equations (ODE-Nets; Chen et al., 2018) for system identification. This is achieved by expressing their dynamics as a truncated series of Legendre polynomials. The series coefficients, as well as the network weights, are computed by minimizing the weighted sum of the loss function and the violation of the ODE-Net dynamics. The problem is solved by coordinate descent that alternately minimizes, with respect to the coefficients and the weights, two unconstrained sub-problems using standard backpropagation and gradient methods. The resulting optimization scheme is fully time-parallel and results in a low memory footprint. Experimental comparison to standard methods, such as backpropagation through explicit solvers and the adjoint technique (Chen et al., 2018), on training surrogate models of small and medium-scale dynamical systems shows that it is at least one order of magnitude faster at reaching a comparable value of the loss function. The corresponding testing MSE is one order of magnitude smaller as well, suggesting generalization capabilities increase.

## 1 Introduction

Neural Ordinary Differential Equations (ODE-Nets; Chen et al., 2018) can learn latent models from observations that are sparse in time. This property has the potential to enhance the performance of neural network predictive models in applications where information is sparse in time and it is important to account for exact arrival times and delays. In complex control systems and model-based reinforcement learning, planning over a long horizon is often needed, while high frequency feedback is necessary for maintaining stability (Franklin et al., 2014). Discrete-time models, including RNNs (Jain & Medsker, 1999), often struggle to fully meet the needs of such applications due to the fixed time resolution. ODE-Nets have been shown to provide superior performance with respect to classic RNNs on time series forecasting with sparse training data. However, learning their parameters can be computationally intensive. In particular, ODE-Nets are memory efficient but time inefficient. In this paper, we address this bottleneck and propose a novel alternative strategy for system identification.

**Summary of contributions.** We propose SNODE, a compact representation of ODE-Nets for system identification with full state information that makes use of a higher-order approximation of its states by means of Legendre polynomials. This is outlined in Section 4. In order to find the optimal polynomial coefficients and network parameters, we develop a novel optimization scheme, which does not require to solve an ODE at each iteration. The resulting algorithm is detailed in Section 3 and is based on backpropagation (Linnainmaa, 1970; Werbos, 1981; Lecun, 1988) and automatic differentiation (Paszke et al., 2017). The proposed method is fully parallel with respect to time and its approximation error reduces exponentially with the Legendre polynomial order (Canuto et al., 1988).

**Summary of numerical experiments.** In Section 5, our method is tested on a 6-state vehicle problem, where it is at least one order or magnitude faster in each optimizer iteration than explicit and adjoint methods, while convergence is achieved in a third of the iterations. At test time, the MSE is reduced by one order of magnitude. In Section 6, the method is used for a 30-state system consisting

---

[*]Corresponding author.

of identical vehicles, coupled via a known collision avoidance policy. Again, our method converges in a third of the iterations required by backpropagation though a solver and each iteration is 50x faster than the fastest explicit scheme.

## 2 NEURAL ORDINARY DIFFERENTIAL EQUATIONS

The minimization of a scalar-valued loss function that depends on the output of an ODE-Net can be formulated as a general constrained optimization problem:

$$
\min_{\theta \in \mathbb{R}^m} \int_{t_0}^{t_1} L(t, x(t)) \, dt, \tag{1}
$$
$$
\text{s.t.} \quad \dot{x}(t) = f(t, x(t), u(t); \theta),
$$
$$
x(t_0) = x_0,
$$

where $x(t) \in \mathbb{X}$ is the state, $u(t) \in \mathbb{U}$ is the input, the loss and ODE functions $L$ and $f$ are given, and the parameters $\theta$ have to be learned. The spaces $\mathbb{X}$ and $\mathbb{U}$ are typically Sobolev (e.g. Hilbert) spaces expressing the smoothness of $x(t)$ and $u(t)$ (see Section 8). Equation (1) can be used to represent several inverse problems, for instance in machine learning, estimation, and optimal control (Stengel, 1994; Law et al., 2015; Ross, 2009). Problem (1) can be solved using gradient-based optimization through several time-stepping schemes for solving the ODE. (Chen et al., 2018; Gholami et al., 2019) have proposed to use the adjoint method when $f$ is a neural network. These methods are typically relying on *explicit* time-stepping schemes (Butcher & Wanner, 1996). Limitations of these approaches are briefly summarized:

**Limitations of backpropagation through an ODE solver.**    The standard approach for solving this problem is to compute the gradients $\partial L / \partial \theta$ using backpropagation through a discrete approximation of the constraints, such as Runge-Kutta methods (Runge, 1895; Butcher & Wanner, 1996) or multi-step solvers (Raissi et al., 2018). This ensures that the solution remains feasible (within a numerical tolerance) at each iteration of a gradient descent method. However, it has several drawbacks: 1) the memory cost of storing intermediate quantities during backpropagation can be significant, 2) the application of implicit methods would require solving a nonlinear equation at each step, 3) the numerical error can significantly affect the solution, and 4) the problem topology can be unsuitable for optimization (Petersen et al., 2018).

**Limitations of adjoint methods.**    ODE-Nets (Chen et al., 2018) solve (1) using the adjoint method, which consists of simulating a dynamical system defined by an appropriate augmented Hamiltonian (Ross, 2009), with an additional state referred to as the adjoint. In the backward pass the adjoint ODE is solved numerically to provide the gradients of the loss function. This means that intermediate states of the forward pass do not need to be stored. An additional step of the ODE solver is needed for the backward pass. This suffers from a few drawbacks: 1) the dynamics of either the hidden state or the adjoint might be unstable, due to the symplectic structure of the underlying Hamiltonian system, referred to as the curse of sensitivity in (Ross, 2009); 2) the procedure requires solving a differential algebraic equation and a boundary value problem which is complex, time consuming, and might not have a solution (Ross & Karpenko, 2012).

**Limitations of hybrid methods.**    ANODE (Gholami et al., 2019) splits the problem into time batches, where the adjoint is used, storing in memory only few intermediate states from the forward pass. This allows to improve the robustness and generalization of the adjoint method. A similar improvement could be obtained using reversible integrators. However, its computational cost is of the same order of the adjoint method and it does not offer further opportunities for parallelization.

## 3 RELAXATION OF THE SUPERVISED LEARNING PROBLEM

Our algorithm is based on two ingredients: i) the discretization of the problem using spectral elements leading to SNODE, detailed in Section 4, and ii) the relaxation of the ODE constraint from (1), enabling efficient training through backpropagation. The latter can be applied directly at the continuous level and significantly reduces the difficulty of the optimization, as shown in our examples.

The problem in (1) is split into two smaller subproblems: one finds the trajectory $x(t)$ that minimizes an unconstrained relaxation of (1). The other trains the network weights $\theta$ such that the trajectory becomes a solution of the ODE. Both are addressed using standard gradient descent and backpropagation. In particular, a fixed number of ADAM or SGD steps is performed for each problem in an alternate fashion, until convergence. In the following, the details of each subproblem are discussed.

**Step 0: Initial trajectory.**    The initial trajectory $x(t)$ is chosen by solving the problem

$$\min_{x(t)\in\mathbb{X}} \int_{t_0}^{t_1} L(t, x(t))\, dt, \tag{2}$$
$$\text{s.t.} \quad x(t_0) = x_0,$$

If this problem does not have a unique solution, a regularization term is added. For a quadratic loss, a closed-form solution is readily available. Otherwise, a prescribed number of SGD iterations is used.

**Step 1: Coordinate descent on residual.**    Once the initial trajectory $x^\star(t)$ is found, $\theta$ is computed by solving the *unconstrained* problem:

$$\min_{\theta\in\mathbb{R}^m} \int_{t_0}^{t_1} R(t, x^\star(t), \theta)\, dt, \quad R(t, x(t), \theta) := \|\dot{x}(t) - f(t, x(t), u(t); \theta)\|^2 \tag{3}$$

If the value of the residual at the optimum $\theta^*$ is smaller than a prescribed tolerance, then the algorithms stops. Otherwise, steps 1 and 2 are iterated until convergence.

**Step 2: Coordinate descent on relaxation.**    Once the candidate parameters $\theta^\star$ are found, the trajectory is updated by minimizing the *relaxed objective*:

$$\min_{x(t)\in\mathbb{X}} \int_{t_0}^{t_1} \gamma L(t, x(t)) + R(t, x(t), \theta^\star)\, dt, \tag{4}$$
$$\text{s.t.} \quad x(t_0) = x_0.$$

**Discussion.**    The proposed algorithm can be seen as an alternating coordinate gradient descent on the relaxed functional used in problem (4), i.e., by alternating a minimization with respect to $x(t)$ and $\theta$. If $\gamma = 0$, multiple minima can exist, since each choice of the parameters $\theta$ would induce a different dynamics $x(t)$, solution of the original constraint. For $\gamma \neq 0$, the loss function in (4) trades-off the ODE solution residual for the data fitting, providing a unique solution. The choice of $\gamma$ implicitly introduces a satisfaction tolerance $\epsilon(\gamma)$, i.e., similar to regularized regression (Hastie et al., 2001), implying that $\|\dot{x}(t) - f(t, x(t); \theta)\| \leq \epsilon(\gamma)$. Concurrently, problem (3) reduces the residual.

## 4 SNODE – HIGH-ORDER DISCRETIZATION OF THE RELAXED PROBLEM

In order to numerically solve the problems presented in the previous section, a discretization of $x(t)$ is needed. Rather than updating the values at time points $t_i$ from the past to the future, we introduce a compact representation of the complete discrete trajectory by means of the spectral element method.

**Spectral approximation.**    We start by representing the scalar unknown trajectory, $x(t)$, and the known input, $u(t)$, as truncated series:

$$x(t) = \sum_{i=0}^{p} x_i \psi_i(t), \quad u(t) = \sum_{i=0}^{z} u_i \zeta_i(t), \tag{5}$$

where $x_i, u_i \in \mathbb{R}$ and $\psi_i(t), \zeta_i(t)$ are sets of given basis functions that span the spaces $\mathbb{X}_h \subset \mathbb{X}$ and $\mathbb{U}_h \subset \mathbb{U}$. In this work, we use orthogonal Legendre polynomials of order $p$ (Canuto et al., 1988) for $\psi_i(t)$, where $p$ is a hyperparameter, and the cosine Fourier basis for $\zeta_i(t)$, where z is fixed.

**Collocation and quadrature.** In order to compute the coefficients $x_i$ of (5), we enforce the equation at a discrete set $\mathbb{Q}$ of collocation points $t_q$. Here, we choose $p+1$ Gauss-Lobatto nodes, which include $t = t_0$. This directly enforces the initial condition. Other choices are also possible (Canuto et al., 1988). Introducing the vectors of series coefficients $x_I = \{x_i\}_{i=0}^p$ and of evaluations at quadrature points $x(t_\mathbb{Q}) = \{x(t_q)\}_{q \in \mathbb{Q}}$, the collocation problem can be solved in matrix form as

$$x_I = M^{-1} x(t_\mathbb{Q}), \quad M_{qi} := \psi_i(t_q). \tag{6}$$

We approximate the integral (3) as a sum of residual evaluations over $\mathbb{Q}$. Assuming that $x(0) = x_0$, the integrand at all quadrature points $t_\mathbb{Q}$ can be computed as a component-wise norm

$$R(t_\mathbb{Q}, x(t_\mathbb{Q}), \theta) = \|DM^{-1} x(t_\mathbb{Q}) - f(t_\mathbb{Q}, x(t_\mathbb{Q}), u(t_\mathbb{Q}); \theta)\|^2, \quad D_{qi} := \dot{\psi}_i(t_q). \tag{7}$$

**Fitting the input data.** For the case when problem (2) admits a unique a solution, we propose a new direct training scheme, $\delta$-SNODE, which is summarized in Algorithm 1. In general, a least-squares approach must be used instead. This entails computing the integral in (2), which can be done by evaluating the loss function $L$ at quadrature points $t_q$. If the input data is not available at $t_q$, we approximate the integral by evaluating $L$ at the available time points. The corresponding alternating coordinate descent scheme $\alpha$-SNODE is presented in Algorithm 2. In the next sections, we study the consequences of a low-data scenario on this approach.

We use fixed numbers $N_t$ and $N_x$ of updates for, respectively, $\theta$ and $x(t)$. Both are performed with standard routines, such as SGD. In our experiments, we use ADAM to optimize the parameters and an interpolation order $p = 14$, but any other orders and solvers are possible.

---

**Algorithm 1** $\delta$-SNODE training

**Input:** $M, D$ from (6)-(7).
$x_I^* \leftarrow \arg\min_x L(Mx)$
**while** $R(Mx_I^*, \theta^*) > \delta$ **do**
   $\theta^* \leftarrow \text{ADAM}_\theta [R(Mx_I^*, \theta)]$
**end while**
**Output:** $\theta^*$

---

**Algorithm 2** $\alpha$-SNODE training

**Input:** $M, D, \theta^* \in \mathbb{R}^m, x_I^* \in \mathbb{R}^p, \gamma > 0$.
**while** $\gamma L(Mx_I^*) + R(Mx_I^*, \theta^*) > \delta$ **do**
   **for** $i = 0 \dots N_x$ **do**
      $x_I^* \leftarrow \text{SGD}_x [\gamma L(Mx) + R(Mx, \theta^*)]$
   **end for**
   **for** $i = 0 \dots N_t$ **do**
      $\theta^* \leftarrow \text{ADAM}_\theta [R(Mx_I^*, \theta)]$
   **end for**
**end while**
**Output:** $\theta^*$

---

**Ease of time parallelization.** If $R(t_q) = 0$ is enforced explicitly at $q \in \mathbb{Q}$, then the resulting discrete system can be seen as an *implicit* time-stepping method of order $p$. However, while ODE integrators can only be made parallel across the different components of $x(t)$, the assembly of the residual can be done in parallel also across time. This massively increases the parallelization capabilities of the proposed schemes compared to standard training routines.

**Memory cost.** If an ODE admits a regular solution, with regularity $r > p$, in the sense of Hilbert spaces, i.e., of the number of square-integrable derivatives, then the approximation error of the SNODE converges exponentially with $p$ (Canuto et al., 1988). Hence, it produces a very compact representation of an ODE-Net. Thanks to this property, $p$ is typically much lower than the equivalent number of time steps of explicit or implicit schemes with a fixed order. This greatly reduces the complexity and the memory requirements of the proposed method, which can be evaluated at any $t$ via (5) by only storing few $x_i$ coefficients.

**Stability and vanishing gradients.** The forward Euler method is known to have a small region of convergence. In other words, integrating very fast dynamics requires a very small time step, $dt$, in order to provide accurate results. In particular, for the solver error to be bounded, the eigenvalues of the state Jacobian of the ODE need to lie into the circle of the complex plane centered at $(-1, \ 0)$ with radius $1/dt$ (Ciccone et al., 2018; Isermann, 1989). Higher-order explicit methods, such as Runge-Kutta (Runge, 1895), have larger but still limited convergence regions. Our algorithms on the other hand are implicit methods, which have a larger region of convergence than recursive (explicit) methods (Hairer et al., 1993). We claim that this results in a more stable and robust training. This

claim is supported by our experiments. Reducing the time step can improve the Euler accuracy but it can still lead to vanishing or exploding gradients (Zilly et al., 2016; Goodfellow et al., 2016). In Appendix C, we show that our methods do not suffer from this problem.

**Experiments setup and hyperparameters.** For all experiments, a common setup was employed and no optimization of hyperparameters was performed. Time horizon $T = 10s$ and batch size of 100 were used. Learning rates were set to $10^{-2}$ for ADAM (for all methods) and $10^{-3}$ for SGD (for $\alpha$-SNODE). For the $\alpha$-SNODE method, $\gamma = 3$ and 10 iterations were used for the SGD and ADAM algorithms at each epoch, as outlined in Algorithm 2. The initial trajectory was perturbed as $x_0 = x_0 + \xi, \xi \sim U(-0.1, 0.1)$. This perturbation prevents the exact convergence of Algorithm 1 during initialization, allowing to perform the alternating coordinate descent algorithm.

## 5  MODELING OF A PLANAR VEHICLE DYNAMICS

Let us consider the system

$$\dot{\eta} = J(\eta)v, \quad M\dot{v} + d(v) + C(v)v = u, \tag{8}$$

where $\eta, v \in \mathbb{R}^3$ are the states, $u = (F_x, 0, \tau_{xy})$ is the control, $C(v)$ is the Coriolis matrix, $d(v)$ is the (linear) damping force, and $J(\eta)$ encodes the coordinate transformation from the body to the world frame (Fossen, 2011). A gray-box model is built using a neural network for each matrix

$$\hat{J}(\eta) = f_J(\eta; \theta_J), \quad \hat{C}(v) = f_C(v; \theta_C), \quad \hat{d}(v) = f_d(v; , \theta_d).$$

Each network consists of two layers, the first with a $\tanh$ activation. Bias is excluded for $f_C$ and $f_d$. For $f_J$, $\sin(\phi)$ and $\cos(\phi)$ are used as input features, where $\phi$ is the vehicle orientation. When inserted in (8), these discrete networks produce an ODE-Net that is a surrogate model of the physical system. The trajectories of the system and the learning curves are shown in Appendix A.

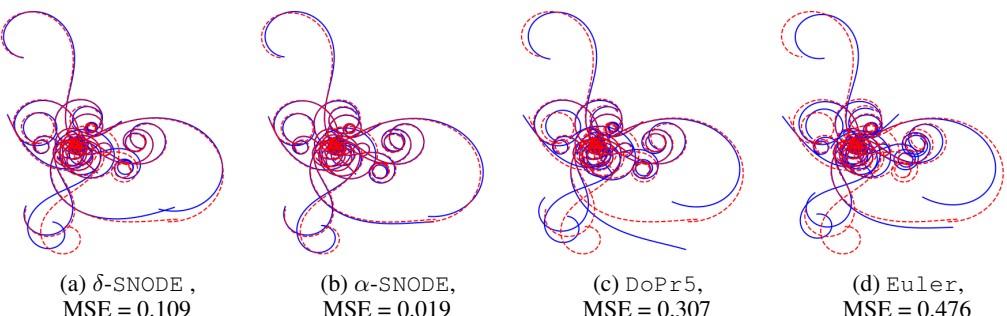

(a) $\delta$-SNODE ,  MSE = 0.109    (b) $\alpha$-SNODE,  MSE = 0.019    (c) DoPr5,  MSE = 0.307    (d) Euler,  MSE = 0.476

Figure 1: **Vehicle dynamics testing in the high-data regime**. Each plot compares the true trajectory (dotted red) with the prediction from the surrogate (solid blue). The time horizon is 5x than in training. The shown methods are: (a)-(b) the proposed $\delta$-SNODE and $\alpha$-SNODE schemes, (c) fifth-order Dormand-Prince, and (d) Explicit Euler. Models trained with our methods have better test MSE and forecast more precisely.

**Comparison of methods in the high-data regime.** In the case of $\delta$-SNODE and $\alpha$-SNODE, only $p + 1$ points are needed for the accurate integration of the loss function, if such points coincide with the Gauss-Lobatto quadrature points. We found that 100 equally-spaced points produce a comparable result. Therefore, the training performance of the novel and traditional training methods were compared by sampling the trajectories at 100 equally-spaced time points. Table 1a shows that $\delta$-SNODE outperforms BKPR-DoPr5 by a factor of 50, while producing a significantly improved generalization. The speedup reduces to 20 for $\alpha$-SNODE, which however yields a further reduction of the testing MSE by a factor of 10, as can be seen in Figure 1.

**Comparison in the low-data regime.** The performance of the methods was compared using fewer time points, randomly sampled from a uniform distribution. For the baselines, evenly-spaced points

Table 1: **Simulation results for the vehicle dynamics example**. The standard backpropagation schemes are denoted by `BKPR`, while the adjoint is denoted by `ADJ`. The Dormand-Prince method of order 5 with adaptive time step is denoted by `DoPr5`, while `Euler` denotes explicit Euler with time step equal to the data sampling time. For `BKPR-DoPr5` we set rtol=$10^{-7}$, atol=$10^{-9}$. Experiments were performed on a i9 Apple laptop with 32GB of RAM. Reported times are per iteration and were averaged over the total number of iterations performed.

(a) **High-data regime**

|  | Training | | | Test |
|---|---|---|---|---|
|  | Time [ms] | # Iter | Final loss | MSE |
| $\delta$-SNODE | **8.3** | 480 | **0.011** | 0.109 |
| $\alpha$-SNODE | 134 | **100** | **0.011** | **0.019** |
| BKPR-Euler | 97 | 1200 | 0.047 | 0.476 |
| BKPR-DoPr5 | 185 | 1140 | **0.011** | 0.307 |
| ADJ-Euler | 208 | 1200 | 0.047 | 0.492 |
| ADJ-DoPr5 | 3516 | 1140 | **0.011** | 0.35 |

(b) **Low-data regime**

|  | Training | | | Test |
|---|---|---|---|---|
|  | % data | # Iter | Final loss | MSE |
| $\alpha$-SNODE | 50 | 140 | 0.010 | 0.029 |
| $\alpha$-SNODE | 25 | 190 | 0.009 | 0.052 |
| BKPR-Euler | 50 | 1200 | 0.163 | 1.056 |
| BKPR-Euler | 25 | 1200 | 0.643 | 3.771 |
| BKPR-DoPr5 | 50 | 1160 | 0.011 | 0.35 |
| BKPR-DoPr5 | 25 | 1180 | 0.011 | 0.291 |

were used. Table 1b shows that $\alpha$-SNODE preserves a good testing MSE, at the price of an increased number of iterations. With only 25% of data, $\alpha$-SNODE is 10x faster than BKPR-DoPr5. Moreover, its test MSE is 1/7 than BKPR-DoPr5 and up to 1/70 than BKPR-Euler, showing that the adaptive time step of DoPr5 improves significantly the baseline but it is unable to match the accuracty of the proposed methods. The adjoint method produced the same results as the backprop ($\pm 2\%$).

## 6  LEARNING A MULTI-AGENT SIMULATION

Consider the multi-agent system consisting of $N_a$ kinematic vehicles:

$$\dot{\eta}_i = J(\eta_i) \tanh \left( w + K_c(\eta_i) + \frac{1}{N_a} \sum_{j \neq i}^{N_a} K_o(\eta_i, \eta_j) \right), \tag{9}$$

where $\eta \in \mathbb{R}^{3N_a}$ are the states (planar position and orientation), $J(\eta_i)$ is the coordinate transform from the body to world frame, common to all agents. The agents velocities are determined by known arbitrary control and collision avoidance policies, respectively, $K_c$ and $K_o$ plus some additional high frequency measurable signal $w = w(t)$, shared by all vehicles. The control laws are non-linear and are described in detail in Appendix B. We wish to learn their kinematics matrix by means of a neural network as in Section 5. The task is simpler here, but the resulting ODE has $3N_a$ states, coupled by $K_0$. We simulate $N_a = 10$ agents in series.

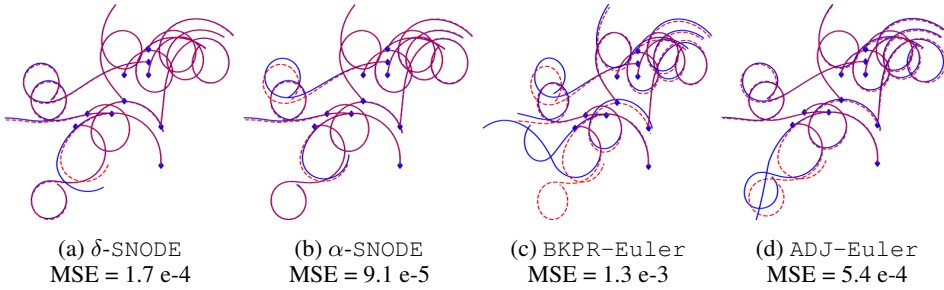

(a) $\delta$-SNODE
MSE = 1.7 e-4

(b) $\alpha$-SNODE
MSE = 9.1 e-5

(c) BKPR-Euler
MSE = 1.3 e-3

(d) ADJ-Euler
MSE = 5.4 e-4

Figure 2: **Multi-agent testing in high-data regime**. Each plot compares the true trajectory (dotted red) with the prediction from the surrogate (solid blue). The time horizon is 4x longer than in training. The shown methods are: (a)-(b) the proposed $\delta$-SNODE and $\alpha$-SNODE schemes, and (c)-(d) Explicit Euler. Models trained with our methods have better test MSE and forecast more precisely.

**Comparison of methods with full and sparse data.**    The learning curves for high-data regime are in Figure 3. For method $\alpha$-SNODE, training was terminated when the loss in (4) is less than $\gamma \bar{L} + \bar{R}$, with $\bar{L} = 0.11$ and $\bar{R} = 0.01$. For the case of 20% data, we set $\bar{L} = 0.01$. Table 2 summarizes results. $\delta$-SNODE is the fastest method, followed by $\alpha$-SNODE which is the best performing. Iteration time of

`BKPR-Euler` is 50x slower, with 14x worse test MSE. `ADJ-Euler` is the slowest but its test MSE is in between `BKPR-Euler` and our methods. Random down-sampling of the data by $50\%$ and $20\%$ (evenly-spaced for the baselines) makes `ADJ-Euler` fall back the most. `BKPR-DoPr5` failed to find a time step meeting the tolerances, therefore they were increased to rtol$= 10^{-5}$, atol$= 10^{-7}$. Since the loss continued to increase, training was terminated at 200 epochs. `ADJ-DoPr5` failed to compute gradients. Test trajectories are in Figure 2. Additional details are in Appendix B.

Table 2: **Simulation results for the multi-agent example**. Experiments were performed on a 2.9 GHz Intel Core i7 Apple laptop with 16GB of RAM. For the $\alpha$-`SNODE` method, each iteration consists of 10 steps of SGD and 10 of ADAM. In high-data regime, $\alpha$-`SNODE` converges in 5x less, 2.6x faster, iterations than Euler with 14x smaller test MSE. Method $\delta$-`SNODE` is even 12x faster, with similar performance. Gains are comparable in low-data regime. Method `BKPR-DoPr5` failed.

| (a) **High-data regime** | | | | |
|---|---|---|---|---|
| | Training | | | Test |
| | Time [ms] | # Iter | Loss | MSE |
| $\delta$-`SNODE` | **106** | **500** | **0.12** | 1.7 e-4 |
| $\alpha$-`SNODE` | 1999 | 180 | **0.12** | **9.1 e-5** |
| `BKPR-Euler` | 5330 | 1200 | 0.177 | 1.3 e-3 |
| `BKPR-DoPr5` | 9580 | **Fail** 200 | 21707 | - |
| `ADJ-Euler` | 8610 | 990 | 0.104 | 5.4 e-4 |
| `ADJ-DoPr5` | - | **Fail** 0 | - | - |

| (b) **Low-data regime** | | | | |
|---|---|---|---|---|
| | Training | | | Test |
| | % data | # Iter | Loss | MSE |
| $\alpha$-`SNODE` | 50 | 160 | 0.22 | 1.5 e-4 |
| $\alpha$-`SNODE` | 20 | 175 | 0.23 | 9.1 e-4 |
| `BKPR-Euler` | 50 | 990 | 0.43 | 2.8 e-3 |
| `BKPR-Euler` | 20 | 990 | 0.645 | 1.6 e-3 |
| `ADJ-Euler` | 50 | 2990 | 0.53 | 9.4 e-3 |
| `ADJ-Euler` | 20 | 1590 | 0.63 | 3.0 e-3 |

**Robustness of the methods.** The use of a high order variable-step method (`DoPr5`), providing an accurate ODE solution, does not however lead to good training results. In particular, the loss function continued to increase over the iterations. On the other hand, despite being nearly 50 times slower than our methods, the fixed-step forward `Euler` solver was successfully used for learning the dynamics of a 30-state system in the *training configuration* described in Appendix B. One should however note that, in this configuration, the gains for the collision avoidance policy $K_o$ (which couples the ODE) were set to small values. This makes the system simpler and more stable than having a larger gain. As a result, if one attempts to train with the *test configuration* from Appendix B, where the gains are increased and the system is more unstable, then backpropagating trough Euler simply fails. Comparing Figures 3 and 4, it can be seen that the learning curves of our methods are unaffected by the change in the gains, while `BKPR-Euler` and `ADJ-Euler` fail to decrease the loss.

# 7 RELATED WORK

**RNN training pathologies.** One of the first RNNs to be trained successfully were LSTMs (Greff et al., 2017), due to their particular architecture. Training an arbitrary RNN effectively is generally difficult as standard RNN dynamics can become unstable or chaotic during training and this can cause the gradients to explode and SGD to fail (Pascanu et al., 2012). When RNNs consist of discretised ODEs, then stability of SGD is intrinsically related to the size of the convergence region of the solver (Ciccone et al., 2018). Since higher-order and implicit solvers have larger convergence region (Hairer

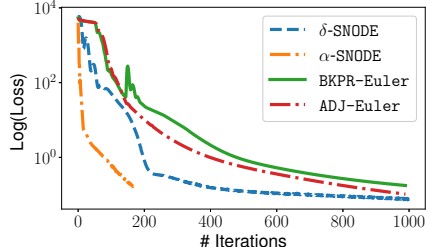
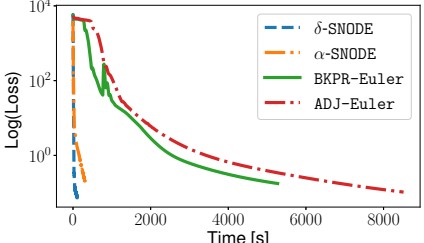

Figure 3: **Multi-agent learning using different methods**. Training loss vs. iterations (left) and execution time (right). Our methods converge one order of magnitude faster and to a lower loss. Note that changing the simulation parameters can make Euler unstable (see Figure 4).

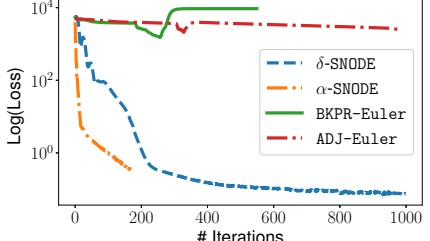 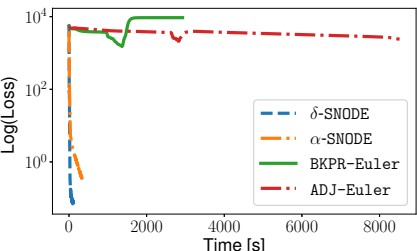

Figure 4: **Multi-agent learning with increased controller gains using different methods**. Training loss vs. iterations (left) and execution time (right). Our methods remain stable and their learning curves are the same as in Figure 3. `BKPR-Euler` and `ADJ-Euler` are not able to train in this configuration. The proposed algorithms are inherently more stable and robust than the baselines.

et al., 1993), following (Pascanu et al., 2012) it can be argued that our method has the potential to mitigate instabilities and hence to make the learning more efficient. This is supported by our results.

**Unrolled architectures.** In (Graves, 2016), an RNN has been used with a stopping criterion, for iterative estimation with adaptive computation time. Highway (Srivastava et al., 2015) and residual networks (He et al., 2015) have been studied in (Greff et al., 2016) as unrolled estimators. In this context, (Haber & Ruthotto, 2017) treated residual networks as autonomous discrete-ODEs and investigated their stability. Finally, in (Ciccone et al., 2018) a discrete-time non-autonomous ODE based on residual networks has been made explicitly stable and convergent to an input-dependant equilibrium, then used for adaptive computation.

**Training stable ODEs.** In (Haber & Ruthotto, 2017; Ciccone et al., 2018), ODE stability conditions where used to train unrolled recurrent residual networks. Similarly, when using our method on (Ciccone et al., 2018) ODE stability can be enforced by projecting the state weight matrices, $A$, into the Hurwitz stable space: i.e. $A \prec 0$. At test time, overall stability will also depend on the solver (Durran, 2010; Isermann, 1989). Therefore, a high order variable step method (e.g. `DoPr5`) should be used at test time in order to minimize the approximation error.

**Dynamics and machine learning.** A physics prior on a neural network was used by (Jia et al., 2018) in the form of a consistency loss with data from a simulation. In (De Avila Belbute-Peres et al., 2018), a differentiable physics framework was introduced for point mass planar models with contact dynamics. (Ruthotto & Haber, 2018) looked at Partial Differential Equations (PDEs) to analyze neural networks, while (Raissi & Karniadakis, 2018; Raissi et al., 2017) used Gaussian Processes (GP) to model PDEs. The solution of a linear ODE was used in (Soleimani et al., 2017) in conjunction with a structured multi-output GP to model patients outcome of continuous treatment observed at random times. (Pathak et al., 2017) predicted the divergence rate of a chaotic system with RNNs.

## 8 SCOPE AND LIMITATIONS

**Test time and cross-validation** At test time, since the future outputs are unknown, an explicit integrator is needed. For cross-validation, the loss needs instead to be evaluated on a different dataset. In order to do so, one needs to solve the ODE forward in time. However, since the output data is available during cross-validation, a corresponding polynomial representation of the form (5) can be found and the relaxed loss (4) can be evaluated efficiently.

**Nonsmooth dynamics.** We have assumed that the ODE-Net dynamics has a regularity $r > p$ in order to take advantage of the exponential convergence of spectral methods, i.e., that their approximation error reduces as $O(h^p)$, where is $h$ is the size of the window used to discretize the interval. However, this might not be true in general. In these cases, the optimal choice would be to use a $hp$-spectral approach (Canuto et al., 1988), where $h$ is reduced locally only near the discontinuities. This is very closely related to adaptive time-stepping for ODE solvers.

**Topological properties, convergence, and better generalization.** There are few theoretical open questions stemming from this work. We argue that one reason for the performance improvement shown by our algorithms is the fact that the set of functions generated by a fixed neural network topology does not posses favorable topological properties for optimization, as discussed in (Petersen et al., 2018). Therefore, the constraint relaxation proposed in this work may improve the properties of the optimization space. This is similar to interior point methods and can help with accelerating the convergence but also with preventing local minima. One further explanation is the fact that the proposed method does not suffer from vanishing nor exploding gradients, as shown in Appendinx C. Moreover, our approach very closely resembles the MAC scheme, for which theoretical convergence results are available (Carreira-Perpinan & Wang, 2014).

**Multiple ODEs: Synchronous vs Asynchronous.** The proposed method can be used for an arbitrary cascade of dynamical systems as they can be expressed as a single ODE. When only the final state of one ODE (or its trajectory) is fed into the next block, e.g. as in (Ciccone et al., 2018), the method could be extended by means of $2M$ smaller optimizations, where $M$ is the number of ODEs.

**Hidden states.** Latent states do not appear in the loss, so training and particularly initializing the polynomial coefficients is more difficult. A hybrid approach is to warm-start the optimizer using few iterations of backpropagation. We plan to investigate a full spectral approach in the future.

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

APPENDIX

## A  ADDITIONAL MATERIAL FOR THE VEHICLE DYNAMICS EXAMPLE

The model is formulated in a concentrated parameter form (Siciliano et al., 2008). We follow the notation of (Fossen, 2011). Recall the system definition:

$$\dot{\eta} = J(\eta)v, \quad M\dot{v} + d(v) + C(v)v = u,$$

where $\eta, v \in \mathbb{R}^3$ are the states, namely, the $x$, and $y$ coordinates in a fixed (world) frame, the vehicle orientation with respect this this frame, $\phi$, and the body-frame velocities, $v_x, v_y$, and angular rate, $\omega$. The input is a set of torques in the body-frame, $u = (F_x, 0, \tau_{xy})$. The Kinematic matrix is

$$J(\eta) = \begin{bmatrix} \cos(\phi) & -\sin(\phi) & 0 \\ \sin(\phi) & \cos(\phi) & 0 \\ 0 & 0 & 1 \end{bmatrix},$$

the mass matrix is

$$M = \begin{bmatrix} m & 0 & 0 \\ 0 & m & 0 \\ 0 & 0 & I \end{bmatrix},$$

where $m$ is the vehicle mass and $I$ represents the rotational intertia. The Coriolis matrix is

$$C(v) = \begin{bmatrix} 0 & -m\omega & 0 \\ m\omega & 0 & 0 \\ 0 & 0 & 0 \end{bmatrix},$$

and the damping force is $d(v) = k_d v$. We set $m = 1$ and $K_d = 1$. The input, $u$, comes from a Fourier series with fundamental amplitude 1.

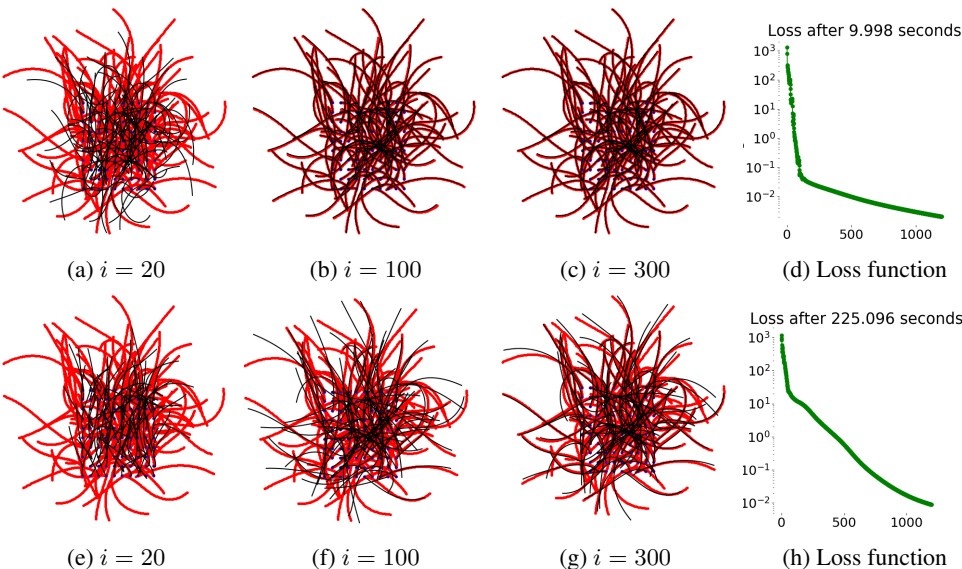

(a) $i = 20$     (b) $i = 100$     (c) $i = 300$     (d) Loss function

(e) $i = 20$     (f) $i = 100$     (g) $i = 300$     (h) Loss function

Figure 5: **Vehicle dynamics learning in high-data regime**. Top row: $\delta$-SNODE method. Bottom row: fifth-order Dormand-Prince method (Butcher & Wanner, 1996) with backpropagation. First three columns: comparison of true trajectories (red) with the prediction from the surrogate (black) at different iterations of the optimization. Last column: loss function at each iteration. $\delta$-SNODE has faster convergence than Euler.

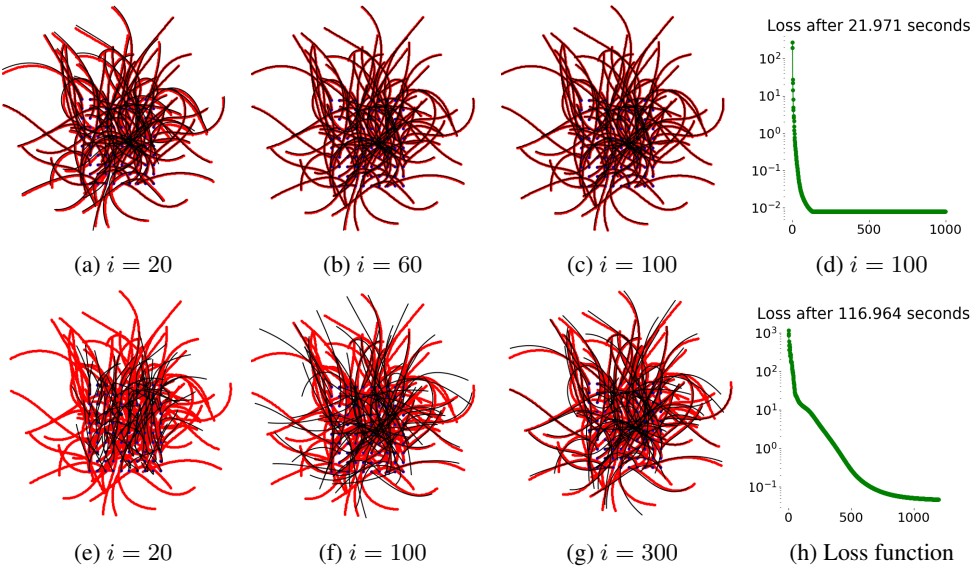

Figure 6: **Vehicle dynamics learning in high-data regime**. Top row: $\alpha$-SNODE method. Bottom row: Explicit Euler method with back-propagation. First three columns: comparison of true trajectories (red) with the prediction from the surrogate (black) at different iterations of the optimization. Last column: value of the loss function at each iteration.

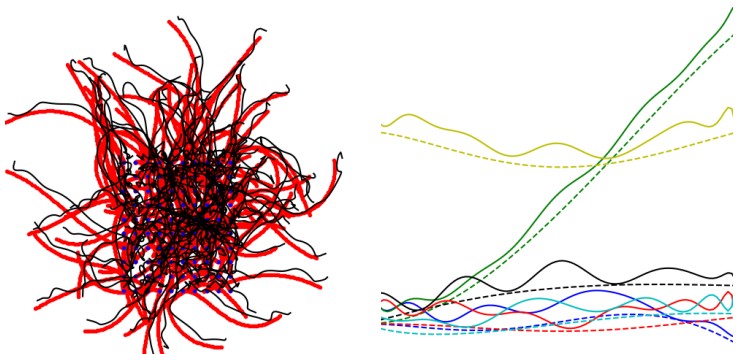

Figure 7: **Vehicle dynamics learning with perturbed input data for the $\alpha$-SNODE method**. Left: comparison of true trajectories (red) with the input data (black). Right: comparison in the time domain for the states of the first sample of the batch.

## B  ADDITIONAL MATERIAL FOR THE MULTI-AGENT EXAMPLE

The multi-agent simulation consists of $N_a$ kinematic vehicles:

$$\dot{\eta}_i = J(\eta_i)v_i, \quad v_i = \tanh\left(w + K_c(\eta_i) + \frac{1}{N_a}\sum_{j\neq i}^{N_a} K_o(\eta_i, \eta_j)\right),$$

where $\eta \in \mathbb{R}^{3N_a}$ are the states for each vehicle, namely, the $x_i, y_i$ positions and the orientation $\phi_i$ of vehicle $i$ in the world frame, while $v_i \in \mathbb{R}^{2N_a}$ are the controls signals, in the form of linear and angular velocities, $\nu_i, \omega_i$. The kinematics matrix is

$$J(\eta_i) = \begin{bmatrix} \cos(\phi_i) & 0 \\ \sin(\phi_i) & 0 \\ 0 & 1 \end{bmatrix}.$$

The agents velocities are determined by known arbitrary control and collision avoidance policies, respectively, $K_c$ and $K_o$. In particular:

$$K_c(\eta_i) = \begin{bmatrix} k_v \\ k_\phi \delta\phi_i \end{bmatrix}, \quad K_o(\eta_i, \eta_j) = \begin{bmatrix} -k_{vo}e^{-\frac{d}{l_s}}e^{-|\delta\phi_{ij}+\pi/2|} \\ k_{\phi o}\delta\phi_{ij} \end{bmatrix},$$

where $\delta\phi_i = \mathtt{atan2}\left((-y_i), (-x_i)\right) - \phi_i$, and $\delta\phi_{ij} = \mathtt{atan2}\left((y_i - y_j), (x_i - x_j)\right) - \phi_i$.

**Training configuration.** We set

$$k_v = 0.05 \quad k_\phi = 0.1, \quad k_{vo} = 0.001, \quad k_{\phi o} = 0.01, \quad l_s = 0.01.$$

The signal $w = w(t)$ is generated by a Fourier series with fundamental amplitude 0.1.

**Test configuration.** We change $K_o(\eta_i, \eta_j)$ to:

$$K_{\text{test}}(\eta_i, \eta_j) = \begin{bmatrix} -k_{vo}e^{-\frac{d}{l_s}} \\ k_{\phi o}\delta\phi_{ij} \end{bmatrix},$$

with $k_{vo} = 0.05$ and $k_{\phi o} = 0.1$. We also set $w = 0$.

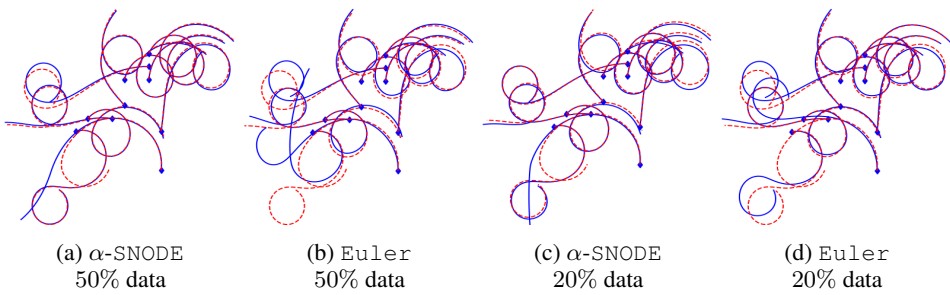

| (a) $\alpha$-SNODE 50% data | (b) Euler 50% data | (c) $\alpha$-SNODE 20% data | (d) Euler 20% data |

Figure 8: **Multi-agent testing in low-data regime**.

## C  GRADIENT ANALYSIS

Consider the classic *discrete-time* RNN:

$$x(t + 1) = f(x(t), u(t); \theta). \tag{10}$$

Then, given a loss $\mathcal{L} = \sum_{t=t_0}^{t_f} L(x(t))$, the following gradients are used during training[1]:

$$\frac{\partial\mathcal{L}}{\partial\theta} = \sum_{t=t_0}^{t_f} \frac{\partial L(x(t))}{\partial x(t)} \frac{\partial x(t)}{\partial\theta}, \tag{11}$$

where, for any $t$, the chain rule gives:

$$\frac{\partial x(t+1)}{\partial\theta} = \frac{\partial f(x(t), u(t); \theta)}{\partial\theta} + J(x(t))\frac{\partial x(t)}{\partial\theta}, \tag{12}$$

$$J(x(t)) = \frac{f(x(t), u(t); \theta)}{\partial x(t)}$$

Iteration of (12) is the main principle of backpropagation through time (Goodfellow et al., 2016). A known fact is that iterating (12) is similar to a geometric series. Therefore, depending on the spectral radius of the Jacobian, $\rho(J(x(t)))$, it can result in vanishing ($\rho < 1$) or exploding ($\rho > 1$) gradients (Zilly et al., 2016).

We can now demonstrate that, by avoiding function compositions, our approach is immune to the exploding gradient problem. In particular, our gradients are fully *time-independent* and their accuracy is *not affected by the sequence length*. Recall the ODE:

$$\dot{x}(t) = f(x(t), u(t); \theta). \tag{13}$$

The following result is obtained:

---

[1]We consider a single batch element: $f$ and $x$ are in $\mathbb{R}^{n_x \times 1}$, where $n_x$ is the state dimensionality.

**Theorem 1.** *Assume the full Jacobian of $f(x(t), u(t); \theta)$ has a finite spectral radius for all $t$. Then, the norms of the gradients used in Algorithm 1 and Algorithm 2 are also finite for all $t$.*

We will prove the theorem for Algorithm 2 since it includes the relevant parts of Algorithm 1.

*Proof.* Given the Legendre basis functions, $\psi_i(t), \ i = 1, \ldots, p$, define the ODE residual as:

$$r(x(t), \theta) = \sum_{i=0}^{p} x_i \dot{\psi}_i(t) - f\left(\sum_{i=0}^{p} x_i \psi_i(t), u(t); \theta\right),$$

for which the residual loss is $R(x(t), \theta) = \|r(x(t), \theta)\|^2$. Then, Algorithm 2 consists of the concurrent minimization of the relaxed loss function:

$$\int_{t_0}^{t_1} \gamma L(x(t))) + R(x(t), \theta^\star) dt,$$

with respect to the coefficients $x_i$ of the Legendre polynomial given $\theta^\star$, and of the residual loss:

$$\int_{t_0}^{t_1} R(x(t), \theta) dt,$$

with respect to $\theta$ given the set of coefficients $x_i^\star, \ i = 1, \ldots, p$. For the residual loss gradients are:

$$\frac{\partial R(x^\star(t), \theta)}{\partial \theta} = -2 \, r(x^\star(t), \theta)^T \frac{\partial f\left(\sum_{i=0}^{p} x_i^\star \psi_i(t), u(t); \theta\right)}{\partial \theta}, \tag{14}$$

where there is no recursion over the previous values $x(t - \tau)$, since the basis functions $\psi_i(t)$ are given and the points $x_i^\star$ are treated as data. By assumption, the Jacobian of $f(x(t), u; \theta)$ has finite singular values. Hence, by standard matrix norm identities (see Chapter 5 of Horn & Johnson (2012)) the gradient of $R(x(t), \theta)$ with respect to $\theta$ has a finite norm for all $t$.

Similarly, the absence of time recursions in the gradient for the $x_i$ update using the relaxed loss also follows trivially from the fact that the coefficients $x_i$ are independent variables, that we assume a given $\theta^\star$, that the basis functions $\psi_i(t)$ are fixed. Then, the claim follows again from the assumption on the Jacobian of $f(x, u; \theta)$. $\square$

Note that the result of Theorem 1 cannot easily be achieved when training ODEs using backpropagation through the solver or the adjoint method unless some gradient conditioning, such as clipping or batch normalization (Ioffe & Szegedy, 2015), is performed *after* applying $f$. On the other hand, our result relies only on the gradient of $f$ being *finite*. This is trivial for a shallow $f$ and, for a very deep $f$, the methods in (Ioffe & Szegedy, 2015; Srivastava et al., 2015; Ciccone et al., 2018; He et al., 2015) can be applied just *inside* the architecture of $f$, if necessary. This fundamental difference with standard RNN training allows for $f$ to have *unrestricted* Jacobian magnitudes which are needed to effectively model instabilities and long-short term dynamics, similar to LSTMs (Greff et al., 2017).

