# OpenReview forum: "SNODE: Spectral Discretization of Neural ODEs for System Identification"
_ICLR.cc/2020/Conference — Accept (Poster)_

### Official Review · AnonReviewer3 · 2019-10-21
**Official Blind Review #3**

**Rating:** 8

**Review:**

TITLE
SNODE: Spectral Discretization of Neural ODEs for System Identification

REVIEW SUMMARY
Exceptionally clear and well written paper that demonstrates a strong improvement on an important problem. Novel, timely, and of broad interest.

PAPER SUMMARY
The paper presents a new method for estimating parameters in a neural ODE based on a polynomial representation of trajectories and alternating updates of trajectories and neural net parameters.

CLARITY
The presentation is exceptionally clear.

ORIGINALITY
To my knowledge the proposed method is novel.

SIGNIFICANCE
The paper demonstrates a strong and practically significant improvement in learning, and I expect the results will be of interest to everybody working in this area.

FURTHER COMMENTS

In eq. 2, "X" (blackboard typeface) is not defined. At this point, it is not clear how x(t) is represented.

"trades-off" -> trades off



**Experience Assessment:**

I have read many papers in this area.

**Review Assessment: Checking Correctness Of Derivations And Theory:**

I assessed the sensibility of the derivations and theory.

**Review Assessment: Checking Correctness Of Experiments:**

I assessed the sensibility of the experiments.

**Review Assessment: Thoroughness In Paper Reading:**

I read the paper at least twice and used my best judgement in assessing the paper.

---

> ### Author Response · Authors · 2019-11-07
> **Reply to Review #3**
>
> We thank the Reviewer for the very positive feedback. With regard to the question concerning  X, it is a function space, typically a Sobolev or a Hilbert space, where the solution x(t) is sought. For our method, we would like it to be a Hilbert space H^p, where p is the order of the polynomial, in order to maximize the accuracy of the representation (as also discussed in Section 8). We will make it clearer after equation (2).

---

### Official Review · AnonReviewer2 · 2019-10-22
**Official Blind Review #2**

**Rating:** 6

**Review:**

This work proposes a new approach for the evolution of Neural ODEs for particle systems. The authors suggest to replace the traditional backpropagation through ODEs or the recent adjoint method for backpropagation and instead solve the problem as an alternating optimization scheme. In particular it is suggested that using spectral methods (Legendre's polynomials) first compute a minimizer of the trajectory (optimizing a trajectory x(t)) given the Loss/Langrangian (that is based on the data times of training). After an initial trajectory is computed, follow an alternating minimization, where in the first step, minimize the discrepancy between the network (that describes the time change of the ODE) and the time derivative of the current trajectory. In the second step, re-compute the trajectory with the new updated network and modified lagrangian to update the network parameters again. This two step optimization is applied back and forth until a residual condition is reached, i.e. the loss is small. The network's parameters in the two stages are optimized via SGD and ADAM respectively. Further, to perform the required numerical integration in each step the authors apply Gaussian quadrature and turn the overall optimization scheme into the repeating application of a finite number of gradient updates in an alternating fashion (as has become popular in recent Deep Learning approaches e.g. GANs). The authors test the new method on different particle systems' trajectories and observe a numerical speed-up and improved accuracy as compared with previous approaches.

Admittingly, I am not an expert in Neural ODEs and am not too familiar with the literature investigating neural networks for modelling differential equations and control. The paper, however is well written and the presentation is very nice in my opinion. It is hard for me to judge how original some of the ideas presented but from my perspective they seem quite solid.

Overall, I am currently voting for weak accept, for solid presentation and content, but with with the following problems:

It seems that Neural ODEs are most beneficial, over the alternatives, when used with irregular data times and or sparse number of time points. This is a point that is mostly missing in the discussion and the experimental section, from my understanding the experimental section uses equally spaced intervals. If this is the case, I do not find them sufficient and I would hope to see how does this method perform when trained with sparser and or irregular time points. Currently the method presented is illustrated as an effective algorithm for noisy ODE solvers. For this reason, I am also wondering whether this is also the right venue to present this interesting work.

Other small things:
In the second sentence
"ODE-Nets have been shown to provide superior performance with respect to classic
RNNs on time series forecasting with sparse training data."
It could be nice to provide a reference illustrating the improved performance of Neural ODEs over RNNS on a time series forecasting task.


**Experience Assessment:**

I do not know much about this area.

**Review Assessment: Checking Correctness Of Derivations And Theory:**

I carefully checked the derivations and theory.

**Review Assessment: Checking Correctness Of Experiments:**

I carefully checked the experiments.

**Review Assessment: Thoroughness In Paper Reading:**

I read the paper thoroughly.

---

> ### Author Response · Authors · 2019-11-07
> **Reply to Review #2**
>
> We thank the Reviewer for the positive feedback. With regards to the main concern raised, namely the regular / irregular time sampling of the data, we apologize that we have not made clear that the results in the low-data / sparse regime were performed by randomly sampling the input data, using a uniform distribution over the entire time interval [0,T] for our methods. For the baselines, we used equally-spaced points. We will make this clearer in the final version.

---

> ### Author Response · Authors · 2019-11-07
> **Reply to Review #2 - part 2**
>
> Concerning the performance of Neural ODEs vs. RNNs on time series forecasting, this is reported in Section 5.1 of [Chen et. al, 2018] (see Table 2). If required by the Reviewer, we will include the citation right after the claim.

---

### Official Review · AnonReviewer4 · 2019-11-02
**Official Blind Review #4**

**Rating:** 6

**Review:**

This work extends prior work on Neural ODEs.  From what I understand, the Neural ODE approach builds off the idea of representing the sequence of transformations of a hidden state (in residual nets, RNNs, etc.) as an ODE parameterized by a neural network.   In the original paper, the network is optimized via gradients calculated by the adjoint sensitivity method.  This paper puts forth the following contributions: a compact representation of the state transition function as a combination of Legendre polynomials, and an optimization scheme whose error is tied to the polynomial order and whose structure lends itself easily to parallelization.  The authors also demonstrate their model on an experiment on planar vehicle dynamics, in which their model is shown to have improved predictive quality and efficiency.

I am inclined to accept this paper, due to its various contributions on speed and performance, with the caveat for some clarifications on how the experiments were conducted and compared.  Given these clarifications in an author response, I would be willing to increase the score.

Pros of the paper:
	1) Trajectory predictions on the two planar vehicle dynamics experiments was impressive.
	2) The proposed representation of the state transition dynamics is indeed more memory-efficient, and its approximation error is modulatable by the hyperparameter order \p.
	3) Speedup due to parallelization is substantial.
Cons of the paper:
	1) The experiments did not display a proper comparison against the hybrid method mentioned in Section 1.  The experiments also did not compare against adjoint methods in the multi-agent example, or in the low-data regime for the single-agent example.  Instead, the experiments mostly highlighted the problems with direct backpropagation through the ODE solver, which is already well-known to have issues in robustness and stability.  While it is nice to have empirical results that showcase this, a more comprehensive comparison against current adjoint methods would be more interesting, especially in the multi-agent example.
	2) It slightly detracts from the cleanliness of the story that we must first create an initial trajectory, before performing our coordinate descent.

Questions and Points of Confusion:
	1) What intuits the choices of the Legendre polynomial as your set of basis functions, and the Gauss-Lobatto scheme to select collocation points, instead of alternative candidates?
	2) In Section 5, it was mentioned that "100 equally-spaced points produce a comparable result" to the Gauss-Lobatto quadrature points.  Furthermore, it was mentioned that these evenly-spaced collocation points were used for experiments - was this the case for all experiments?  If so, then what purpose does Gauss-Lobatto play in the paper?
	3) In Figure 1, were the DoPr5 and Euler plots shown for the backpropagation or adjoint method?  If it was the backpropagation method, the plots for the adjoint method would be highly interesting to show.
	4) From what I understand from Section 4, when we perform step 0 and step 2 to update the trajectory, we simply optimize the coefficients \x_i and \u_i directly to minimize the current objective as both \x(t) and \u(t) are represented as a combination of Legendre polynomials.  However, in Equation 8, for the planar vehicle dynamics, \u is deterministically generated from the current states and network weights.  It makes sense to add structure to \u, as we need a way to make sure the inputs can indeed generate the trajectory of \x.  Does this mean the spectral method is not performed for \u as was detailed in Equation 5?
	5) I am confused about how the gray-box models are built in Section 5.  It states that "For \f_J, sin(\phi) and cos(\phi) are used as input features, where \phi is the vehicle orientation."  Does that mean that only \phi from \eta is passed in as an input, both sin(\phi) and cos(\phi) are passed in as inputs, or the entire current \eta is passed in as input?  And is the output a new \eta, which is then structured into the 3x3 matrix \J(\eta) in Appendix A?  Or does it simply output the matrix directly for the given value of \phi.  I am confused because it is written that \J(\eta) is equal to \f_J(\eta;\theta_J), but then in the appendix it is written that \J(\eta) is equal to the matrix - so where is the network?  This confusion extends to the other gray-box models \C(v) and \d(v).
	6) It is written in Equation 3 that the residual also takes in the input \u(t).  However, in the experiments, namely Equation 8, it does not appear like \u is used to calculate the residual at all.
	7) What motivated the use of the planar vehicle dynamics experiment to showcase your model?  Were there other baselines or benchmarks you considered or attempted?


**Experience Assessment:**

I do not know much about this area.

**Review Assessment: Checking Correctness Of Derivations And Theory:**

I assessed the sensibility of the derivations and theory.

**Review Assessment: Checking Correctness Of Experiments:**

I assessed the sensibility of the experiments.

**Review Assessment: Thoroughness In Paper Reading:**

I read the paper thoroughly.

---

> ### Author Response · Authors · 2019-11-07
> **Reply to Review #4 - part 1**
>
> We thank the Reviewer for the positive feedback.
>
> Major points:
> 1) Following the Reviewer's comments, we performed further comparisons against the adjoint method. In particular, in the low-data regime for the single-agent example the generalization results and trajectories are identical to backpropagation. We are also running the high-data regime for the multi-agent example. We will add the full results in the table of the final version of the paper. If the Reviewer deems it necessary, we could add the corresponding figures to the appendix, although they are identical to backpropagation. For the hybrid method [Gholami et al, 2019], we recognize it could offer an improvement in stability and accuracy also for these examples with respect to the adjoint and backprop methods. We will make this clearer in Section 2. We have not investigated this method experimentally since its computational cost is the same as that of the adjoint method and our main focus was on speed and parallelization.
> 2) We agree with the Reviewer that the choice of initial trajectory plays a critical role. We note that however this is in general true for all optimization problems. For the proposed algorithm, there are several possible options to do this. A trivial approach would be to repeat the known initial condition $x_0$ over the entire time interval. A second approach would be to integrate once the ODE in time using the initial network weights. This choice would yield a zero residual but may produce weights that are similar to standard backpropagation. A third and preferred approach is to perform a fit of the data according to some user-prescribed criterion. This is the approach that we used. When weights exist such that the ODE can follow this trajectory, the delta method is obtained. The role of the alpha method is therefore to slightly correct the desired trajectory so that it can be approximated by the ODE.

---

> ### Author Response · Authors · 2019-11-07
> **Reply to Review #4 - part 2**
>
> 1) Among the possible choices presented in ref. [Canuto et al.], we selected the combination of Legendre polynomials and Gauss-Lobatto points for ease of implementation. In particular, the Gauss-Lobatto nodes include the point , where the initial condition is specified. This makes is easier to directly replace the ODE at  with the initial condition, rather than adding one equation to the residual. Moreover, Lagrange polynomials provide a simple recursive formula for the computation of derivatives and produce better conditioning of the system matrices compared with Lagrange polynomials. The drawback is that the resulting mass matrix M is non-diagonal. However, for our case M is small and its inverse can be stored directly. We also remark that any other combination of polynomials and quadrature methods is possible in our framework. We will make this clearer in the paper.
> 2) There are two parts of the algorithms where quadrature points are needed: for the computation of the residual and of the loss. For the residual, we always use the Gauss-Lobatto points. For the loss, which requires evaluating the data at the quadrature points, we have compared equally-spaced and Gauss-Lobatto points in the high-data regime. Since the results are very similar, as expected for a smooth integrand, Gauss-Lobatto provide a much more efficient option. In the low/sparse regime, we have used randomly sampled points for our methods and equally-spaced for the baselines. We will make this clearer in the paper.
> 3) See point 1. of the major points above.
> 4) In equation (8), the state variables are v and . The parameter u corresponds to the representation of the input data (e.g. torques trajectories represented through Legendre polynomials computed in the initialization phase). The forward model then solves for v and , given u. In the experiments, however, we use u(t) as per equation (5), with the following differences: (i) we used a truncated Fourier basis instead of Legendre polynomials and (ii) rather than optimizing for the coefficients , we generate them using a random distribution and keep them fixed during training. This is written in appendix A and B, but we will make it clearer in the main text. We use Fourier basis in order to generate the data for the example. Therefore, in this case there was no need to train an additional polynomial for . In the general case, one would train a representation of u given the input data, as written in equation (5).
> 5) We recognize that in the unnumbered equation below (8), the networks are defined with the same notation as for the simulation generating the data (in appendix). To clarify, we train fully-connected networks using  and  features for  (as written in the paper below the unnumbered equation). The output of  has dimension 9, which is then reshaped to the 3x3 tensor . The same holds true for the other two tensors., with the only difference being that the bias is deactivated for the networks  and . We will make sure that the notation is different for the surrogate model and for the simulation. We propose to use  for the surrogate.
> 6) See point 4. above.
> 7) The choice for the examples comes from the need in modeling and control to obtain reliable forecast for both the short- and long-term dynamics. This is a particularly sensitive issue in robotics and automation as errors can lead to instabilities and unsafe decisions.

---

### Public Comment · ~Yiping_Lu1 · 2019-09-27
**Related works**

Congrats on your work and I really enjoy reading it.
I'm writting the comment to introduce some of our related works on discretization Neural ODEs and system identification
Lu Y, Zhong A, Li Q, et al. Beyond finite layer neural networks: Bridging deep architectures and numerical differential equations[J]. arXiv preprint arXiv:1710.10121, 2017.
Long Z, Lu Y, Ma X, et al. PDE-net: Learning PDEs from data[J]. arXiv preprint arXiv:1710.09668, 2017.


Also these early papers also aim to connect ODEs and deep learning
Weinan, E. "A proposal on machine learning via dynamical systems." Communications in Mathematics and Statistics 5.1 (2017): 1-11.
Li, Qianxiao, et al. "Maximum principle based algorithms for deep learning." The Journal of Machine Learning Research 18.1 (2017): 5998-6026.
Weinan, E., Jiequn Han, and Qianxiao Li. "A mean-field optimal control formulation of deep learning." Research in the Mathematical Sciences 6.1 (2019): 10.
Haber, Eldad, and Lars Ruthotto. "Stable architectures for deep neural networks." Inverse Problems 34.1 (2017): 014004.
Chang, Bo, et al. "Multi-level residual networks from dynamical systems view." arXiv preprint arXiv:1710.10348 (2017).
Ruthotto, Lars, and Eldad Haber. "Deep neural networks motivated by partial differential equations." arXiv preprint arXiv:1804.04272 (2018).


And these works are working on the conjugate method
Li, Qianxiao, et al. "Maximum principle based algorithms for deep learning." The Journal of Machine Learning Research 18.1 (2017): 5998-6026.
Dinghuai Zhang*, Tianyuan Zhang*,Yiping Lu*, Zhanxing Zhu, Bin Dong. "You Only Propagate Once: Painless Adversarial Training Using Maximal Principle." (*equal contribution) 33rd Annual Conference on Neural Information Processing Systems 2019(NeurIPS2019).
Li Q, Hao S. An optimal control approach to deep learning and applications to discrete-weight neural networks[J]. arXiv preprint arXiv:1803.01299, 2018.

---

### Author Response · Authors · 2019-11-08
**Revision 1 uploaded**

We have uploaded the first revision, containing most of discussed amendments from the Reviews. We are running few more experiments with the adjoint method baseline on the multi-agent example. Initial experiments are already included in this revision.

---

### Author Response · Authors · 2019-11-14
**Final revision uploaded**

We have uploaded the final revision, containing the amendments suggested by the Reviewers and the results of the additional experiments using the adjoint method for the multi-agent example. Further details concerning all of the points raised in each review can be found in the comments that we posted last week below.

---

### Decision · Program_Chairs · 2019-12-19

**Decision:**

Accept (Poster)

**Comment:**

This work proposes using spectral element methods to speed up training of ODE Networks for system identification. The authors utilize truncated series of Legendre polynomials to analyze the dynamics and then conduct experiments that shows their proposed scheme achieves an order of magnitude improvement in training speed compared to baseline methods. Reviewers raised some concerns (e.g. empirical comparison against adjoint methods in the multi-agent example) or asked for clarifications (e.g. details of time sampling of the data). The authors adequately addressed most of these concerns via rebuttal response as well as revising the initial submission. At the end, all reviewers recommended for accept based on contributions of this work on improving training speed of ODE Networks. R4 hopes that some of the additional concerns that are not yet reflected in the current revision, be addressed in the camera ready version.